# Artificial Replay: A Meta-Algorithm for Harnessing Historical Data in Bandits

## Abstract

While standard bandit algorithms sometimes incur high regret, their performance can be greatly improved by "warm starting" with historical data. Unfortunately, how best to incorporate historical data is unclear: naively initializing reward estimates using all historical samples can suffer from *spurious data* and *imbalanced data coverage*, leading to computational and storage issues—particularly in continuous action spaces. We address these two challenges by proposing Artificial Replay, a meta-algorithm for incorporating historical data into *any arbitrary base bandit algorithm*. Artificial Replay uses only a subset of the historical data *as needed* to reduce computation and storage. We provide guarantees that our method achieves equal regret as a full warm-start approach while potentially using only a fraction of the historical data for a broad class of base algorithms that satisfy *independence of irrelevant data* (IIData), a novel property that we introduce. We complement these theoretical results with a case study of $K$-armed and continuous combinatorial bandit algorithms, including on a green security domain using real poaching data, to show the practical benefits of Artificial Replay in achieving optimal regret alongside low computational and storage costs. Across these experiments, we show that Artificial Replay performs well for all settings that we consider, even for base algorithms that do not satisfy IIData.

## 1 Introduction

Multi-armed bandits and their variants are robust models for many real-world problems. Resulting algorithms have been applied to wireless networks (Zuo & Joe-Wong, 2021), COVID testing regulations (Bastani et al., 2021), and conservation efforts to protect wildlife from poaching (Xu et al., 2021). Typical bandit algorithms assume no prior knowledge of the expected rewards of each action, simply taking actions online to address the exploration–exploitation trade-off. However, many real-world applications offer access to *historical data*. For example, in the wildlife conservation setting, we may have access to years of historical patrol records that should be incorporated to learn poaching risk before deploying any bandit algorithm.

There is no consensus on how to optimally incorporate this historical data into online learning algorithms. The naive approach uses the full historical dataset to initialize reward estimates (Shivaswamy & Joachims, 2012), possibly incurring unnecessary and onerous computation and storage costs. These costs are particularly salient in continuous action settings with adaptive discretization, where the number of discretized regions is a direct function of the number of historical samples. If excessive data was collected on poor-performing actions, this *spurious data* with *imbalanced data coverage* would lead us to unnecessarily process and store an extremely large number of fine discretizations in low-performing areas of the action space, even when a significantly coarser discretization would be sufficient to inform us that region is not worth exploring. These two key challenges highlight that *the value of information of the historical dataset may not be a direct function of its size*. Real-world decision makers echo this sentiment: Martin et al. (2017) note that for conservation decisions, more information does not always translate into better actions; time is the resource which matters most.

A natural question one can ask is: *Is there an efficient way (in terms of space, computational, and sample complexity) to use historical data to achieve regret-optimal performance?* For example, many real-world applications of bandit algorithms, such as online recommender systems, may con-

tain historical datasets with millions of data points. Processing these millions of points would require an exceptional amount of upfront computation and storage cost, especially if many of those historical points are no longer relevant; many samples may encode out-of-date data such as old movies or discontinued products.

To this end, we propose ARTIFICIAL REPLAY, a meta-algorithm that modifies *any base bandit algorithm* to harness historical data. ARTIFICIAL REPLAY reduces computation and storage costs by only using historical data on an *as needed* basis. The key intuition is if we could *choose* which samples to include in the historical dataset, a natural approach would be to use a regret-optimal bandit algorithm to guide the sampling. ARTIFICIAL REPLAY builds on this intuition by using historical data as a replay buffer to artificially simulate online actions. Every time the base algorithm picks an action, we first check the historical data for any unused samples from the chosen action. If an unused sample exists, update the reward estimates and continue *without* advancing to the next timestep. Otherwise, sample from the environment, update the estimates using the observion, and continue to the next timestep. While this idea is easiest to understand in the context of the standard $K$-armed bandit problem, we discuss later how this framework naturally extends to other structure and information models, including continuous action spaces with semi-bandit feedback.

Although ARTIFICIAL REPLAY seems to be a natural heuristic to minimize use of historical data, it is not clear how to analyze its regret—specifically how much it loses compared to "full warm-start" (i.e., where the base algorithm is initialized with the full dataset). Surprisingly, however, we prove that under a widely applicable condition, the regret of ARTIFICIAL REPLAY (as a random variable) is distributionally identical to that of a full warm-start approach, while also guaranteeing significantly better time and storage complexity. Specifically, we show a *sample-path coupling*[1] between our AR-TIFICIAL REPLAY approach and the full warm start approach with the same base algorithm, as long as the base algorithm satisfies a novel *independence of irrelevant data* (IIData) assumption. While our goal is not to show regret improvements, this result highlights how ARTIFICIAL REPLAY is a simple approach for incorporating historical data with identical regret to full warm start (approach done in practice) with significantly smaller computational overhead.

Finally, we show the practical benefits of our method by instantiating ARTIFICIAL REPLAY for several broad classes of bandits and evaluating on real-world data. To highlight the breadth of algorithms that satisfy the IIData property, we provide examples of regret-optimal IIData polices for $K$-armed and continuous combinatorial bandits. We use these examples to prove that ARTIFICIAL REPLAY can lead to arbitrary better storage and computational complexity requirements. We close with a case study of combinatorial bandit algorithms for continuous resource allocation in the context of green security domains, using a novel adaptive discretization technique. Across the experiments, we observe concrete gains in storage and runtime using real-world poaching data from the ARTIFICIAL REPLAY framework over a range of base algorithms, including algorithms that do not satisfy IIData such as Thompson sampling and Information Directed Sampling (IDS).

## 1.1 RELATED WORK

Multi-armed bandit problems have a long history in the online learning literature. We highlight the most closely related works below; for more extensive references please see our detailed discussion in Appendix B and Bubeck et al. (2012); Slivkins (2019); Lattimore & Szepesvári (2020).

**Multi-Armed Bandit Algorithms.** The design and analysis of bandit algorithms have been considered under a wide range of models. These algorithms were first studied in the $K$-armed bandit model in Lai & Robbins (1985), where the decision maker has access to a finite set of $K$ possible actions at each timestep. However, numerous follow-up works have considered similar approaches when designing algorithms in continuous action spaces (Kleinberg et al., 2019) and with combinatorial constraints (Chen et al., 2013; Xu et al., 2021; Zuo & Joe-Wong, 2021). Our work provides a framework to modify existing algorithms to harness historical data. Moreover, we also propose a novel algorithm to incorporate adaptive discretization for combinatorial multi-armed bandits for continuous resource allocation, extending the discrete model from Zuo & Joe-Wong (2021).

---

[1]That is, we construct the regret process for both algorithms simultaneously on a joint probability space such that each individual process has the correct marginal, but both processes are equal over each sample path.

**Incorporating Historical Data.**    Several papers have started to investigate how to incorporate historical data into bandit algorithms, starting with Shivaswamy & Joachims (2012) who consider a $K$-armed bandit model where each arm has a dataset of historical pulls. The authors develop a "warm start" UCB algorithm to initialize the confidence bound of each arm based on the full historical data—prior to learning. Bouneffouf et al. (2019) extended similar techniques to models with pre-clustered arms. These techniques were later extended to Bayesian and frequentist linear contextual bandits, where the linear feature vector is initialized based on standard regression over the historical data (Oetomo et al., 2021; Wang et al., 2017). Our work provides a contrasting approach to harnessing historical data in algorithm design: our meta-algorithm can be applied to *any* standard bandit framework and uses the historical data only *as needed*, leading to improved computation and storage gains.

## 2   PRELIMINARIES

We now define the general bandit model and specify the finite-armed and online combinatorial allocation settings that we study in our experiments. See Appendix C for details.

### 2.1   GENERAL STOCHASTIC BANDIT MODEL

We consider a stochastic bandit problem with a fixed action set $\mathcal{A}$. Let $\Re : \mathcal{A} \to \Delta([0,1])$ be a collection of independent and *unknown* reward distributions over $\mathcal{A}$. Our goal is to pick an action $a \in \mathcal{A}$ to maximize $\mathbb{E}[\Re(a)]$, the expected reward, which we denote $\mu(a)$. The optimal reward is:

$$\text{OPT} = \max_{a \in \mathcal{A}} \mu(a) . \tag{1}$$

For now, we do not impose any additional structure on $\mathcal{A}$, which could potentially be discrete, continuous, or encode combinatorial constraints.

**Historical Data.**    We assume that the algorithm designer has access to a historical dataset $\mathcal{H}^{hist} = \{a_j^{\mathcal{H}}, R_j^{\mathcal{H}}\}_{j \in [H]}$ containing $H$ historical points with actions $\{a_j^{\mathcal{H}}\}_{j \in [H]}$ and rewards $R_j^{\mathcal{H}}$ sampled according to $\Re(a_j^{\mathcal{H}})$. We do not make any assumptions on how the historical actions $a_j^{\mathcal{H}}$ are chosen and view them as deterministic and fixed upfront. Our goal is to efficiently incorporate this historical data to improve the performance of a bandit algorithm.

**Online Structure.**    Since the mean reward function $\mu(a)$ is initially unknown, we consider settings where the algorithm interacts with the environment sequentially over $T$ timesteps. At timestep $t \in [T]$, the decision maker picks an action $A_t \in \mathcal{A}$ according to their policy $\pi$. The environment then reveals a reward $R_t$ sampled from the distribution $\Re(A_t)$. The optimal reward OPT would be achieved using a policy with full knowledge of the true distribution. We thus define *regret* as:

$$\text{REGRET}(T, \pi, \mathcal{H}^{hist}) = T \cdot \text{OPT} - \sum_{t=1}^{T} \mu(A_t) . \tag{2}$$

where the dependence on $\mathcal{H}^{hist}$ highlights that $A_t$ can additionally depend on the historical dataset.

### 2.2   FINITE, CONTINUOUS, AND COMBINATORIAL ACTION SPACES

*Finite-Armed Bandit.*    The finite-armed bandit model can be viewed in this framework by considering $K$ discrete actions $\mathcal{A} = [K] = \{1, \dots, K\}$.

*Combinatorial Multi-Armed Bandit for Continuous Resource Allocation (CMAB-CRA).*    A central planner has access to a metric space $\mathcal{S}$ of resources with metric $d_{\mathcal{S}}$. They are tasked with splitting a total amount of $B$ divisible budget across $N$ different resources within $\mathcal{S}$. An action consists of choosing $N$ resources, i.e., $N$ points in $\mathcal{S}$, and allocating the budget among that chosen subset. The feasible space of allocations is $\mathcal{B} = [0, 1]$ and the feasible action space is:

$$\mathcal{A} = \left\{ (\vec{\mathbf{p}}, \vec{\beta}) \in \mathcal{S}^N \times \mathcal{B}^N \mid \sum_{i=1}^{N} \beta^{(i)} \leq B, \quad d_{\mathcal{S}}(\mathbf{p}^{(i)}, \mathbf{p}^{(j)}) \geq \epsilon \quad \forall i \neq j \right\} . \tag{3}$$

The chosen action must satisfy the budgetary constraint (i.e., $\sum_i \beta^{(i)} \leq B$), and the resources must be distinct (aka $\epsilon$-away from each other according to $d_{\mathcal{S}}$ for some $\epsilon > 0$) to ensure the "same" resource is not chosen at multiple allocations. We additionally assume that $\Re$ decomposes independently over the (resource, allocation) pairs, in that $\mu(a) = \sum_{i=1}^{N} \mu(\mathbf{p}^{(i)}, \beta^{(i)})$. Lastly, we assume

---

**Algorithm 1** ARTIFICIAL REPLAY

---
**Require:** Historical dataset $\mathcal{H}^{hist} = \{(a_j^{\mathcal{H}}, R_j^{\mathcal{H}})\}_{j \in [H]}$, base algorithm $\Pi$
  1: Initialize set of used historical data points $\mathcal{H}_1^{on} = \emptyset$, and set of online data $\mathcal{H}_1 = \emptyset$
  2: **for** $t = \{1, 2, \ldots\}$ **do**
  3:     Initialize `flag` to be `True`
  4:     **while** `flag` is `True` **do**
  5:         Pick action $\tilde{A}_t \sim \Pi(\mathcal{H}_t^{on} \cup \mathcal{H}_t)$
  6:         **if** $\tilde{A}_t$ is not contained in $\mathcal{H}^{hist} \setminus \mathcal{H}_t^{on}$ **then**
  7:             Update `flag` to be `False`                    ▷ Finish a full timestep
  8:             Set online action $A_t = \tilde{A}_t$
  9:             Execute action $A_t$ and observe reward $R_t \sim \Re(A_t)$        ▷ Take online sample
 10:             Update $\mathcal{H}_{t+1} = \mathcal{H}_t \cup \{(A_t, R_t)\}$ and $\mathcal{H}_{t+1}^{on} = \mathcal{H}_t^{on}$
 11:         **else**
 12:             Update $\mathcal{H}_t^{on}$ to include one sample for $\tilde{A}_t$ from historical dataset $\mathcal{H}^{hist}$

---

the algorithm observes semi-bandit feedback of the form $(\mathbf{p}_t^{(i)}, \beta_t^{(i)}, R_t^{(i)})_{i \in [N]}$ for each resource and allocation pair sampled according to $\Re(\mathbf{p}_t^{(i)}, \beta_t^{(i)})$. Zuo & Joe-Wong (2021) proposed a discrete model of this problem as a generalization of the works in Dagan & Koby (2018); Lattimore et al. (2014; 2015) specialize it to consider scheduling a finite set of resources to maximize the expected number of jobs finished.

*Extension to Green Security.* The CMAB-CRA model can be used to specify green security domains from Xu et al. (2021) by letting the space $\mathcal{S}$ represent a protected area and letting $\mathcal{B}$ represent the discrete set of patrol resources to allocate, such as number of ranger hours per week, with the total budget $B$ being 40 hours. This formulation generalizes to a more realistic continuous space model of the landscape, instead of the artificial fixed discretization that was considered in prior work consisting of $1 \times 1$ sq. km regions of the park. This also highlights the practical necessity that the chosen resources (here, the patrol locations) are $\epsilon$-far away to ensure sufficient spread. In Section 5, we show that enabling patrol planning at a continuous level can help park rangers more precisely identify poaching hotspots.

## 3   ARTIFICIAL REPLAY FOR HARNESSING HISTORICAL DATA

We propose ARTIFICIAL REPLAY, a meta-algorithm that can be integrated with any base algorithm to incorporate historical data. We later prove that for any base algorithm satisfying *independence of irrelevant data* (IIData), a novel property we introduce, ARTIFICIAL REPLAY has identical regret to an approach which uses the full historical data upfront—showing that our approach reduces computation costs without trading off performance. Additionally, in Appendix E we discuss empirical improvements of ARTIFICIAL REPLAY applied to Thompson Sampling and Information Directed Sampling, two algorithms which do not satisfy IIData.

**Algorithm Formulation.** Any algorithm for online stochastic bandits can be thought of as a function mapping arbitrary ordered histories (i.e., collections of observed $(a, R)$ pairs) to a distribution over actions in $\mathcal{A}$. More specifically, let $\Pi : \mathcal{D} \to \Delta(\mathcal{A})$ be an arbitrary base algorithm where $\mathcal{D}$ denotes the collection of possible histories (i.e., $\mathcal{D} = \cup_{i \geq 0}(\mathcal{A} \times \mathbb{R}_+)$). The policy obtained by a base algorithm $\Pi$ *without incorporating historical data* simply takes the action sampled according to the policy $\pi_t^{\text{IGNORANT}(\Pi)} = \Pi(\mathcal{H}_t)$ where $\mathcal{H}_t$ is the data observed by timestep $t$. In comparison, consider an algorithm $\pi_t^{\text{FULL START}(\Pi)}$ which follows the same policy but uses the *full historical data upfront*, so takes the action sampled according to $\Pi(\mathcal{H}^{hist} \cup \mathcal{H}_t)$.

### 3.1   ARTIFICIAL REPLAY

The ARTIFICIAL REPLAY meta-algorithm incorporates the historical data $\mathcal{H}^{hist}$ into an arbitrary base algorithm $\Pi$, resulting in a policy we denote by $\pi^{\text{ARTIFICIAL REPLAY}(\Pi)}$. See Algorithm 1 for the pseudocode. We let $\mathcal{H}_t^{on}$ be the set of historical datapoints used by the start of time $t$. Initially, $\mathcal{H}_1^{on} = \emptyset$. For an arbitrary timestep $t$, the ARTIFICIAL REPLAY approach works as follows:

Let $\tilde{A}_t \sim \Pi(\mathcal{H}_t^{on} \cup \mathcal{H}_t)$ be the *proposed action* at the start of time $t$. Since we are focused on *simulating* the algorithm with historical data, we break into cases whether or not the current set of unused historical datapoints (i.e., $\mathcal{H}^{hist} \setminus \mathcal{H}_t^{on}$) contains any additional information about $\tilde{A}_t$.

- **No historical data available**: If $\tilde{A}_t$ is not contained in $\mathcal{H}^{hist} \setminus \mathcal{H}_t^{on}$, then the *selected action* is $A_t = \tilde{A}_t$, and we advance to timestep $t + 1$. We additionally set $\mathcal{H}_{t+1}^{on} = \mathcal{H}_t^{on}$.

- **Historical data available**: If $\tilde{A}_t$ is contained in $\mathcal{H}^{hist} \setminus \mathcal{H}_t^{on}$, add that data point to $\mathcal{H}_t^{on}$ and repeat by picking another *proposed action*. We remain at time $t$.

Strikingly, our framework imposes minimal computational and storage overhead on top of existing algorithms, simply requiring a data structure to verify whether $\tilde{A} \in \mathcal{H}^{hist} \setminus \mathcal{H}_t^{on}$, which can be performed with hashing techniques. It is clear that the runtime and storage complexity of ARTIFICIAL REPLAY is no worse than FULL START. We also note that most practical bandit applications incorporate historical data obtained from database systems (e.g. content recommendation systems, wildlife poaching model discussed). This historical data will be stored regardless of the algorithm being employed, and so the key consideration is around computational requirements and not storage.

Additionally, our approach extends naturally to the following different models:

*Continuous Spaces.* The ARTIFICIAL REPLAY framework can be applied in continuous action spaces with discretization-based algorithms. For example, suppose that $\Pi$ wants to select an action $a \in \mathcal{A}$, but the historical data has a sample from $a + \epsilon$, a slightly perturbed point. Discretization-based algorithms avoid precision issues since they map the continuous space to a series of regions which together cover the action set, and run algorithms or subroutines over the discretization. Checking for historical data simply checks for data within the bounds of the chosen discretized action.

*Semi-Bandit Feedback.* ARTIFICIAL REPLAY also naturally extends to combinatorial action sets with semi-bandit feedback where actions are *decomposable*, that is, they can be written as $a = (a_1, \ldots, a_N)$ with independent rewards. Suppose that $\Pi$ wants to select an action $a = (a_1, a_2, \ldots, a_N)$ but the historical data has a sample from $(a_1', a_2, \ldots a_N')$. Even if the combinatorial action $a$ does not appear in its entirety in the historical data, as long as there exists some subcomponent $a_i^{\mathcal{H}}$ (sometimes referred to as "subarm" in combinatorial bandits) in the historical data (e.g., $a_2$) , we can add that subcomponent $a_i^{\mathcal{H}}$ to $\mathcal{H}_t^{on}$ to update the base algorithm.

## 3.2 INDEPENDENCE OF IRRELEVANT DATA AND REGRET COUPLING

It is not immediately clear how to analyze the regret of ARTIFICIAL REPLAY. To enable regret analysis, we introduce a new property for bandit algorithms, *independence of irrelevant data*, which essentially requires that when an algorithm is about to take an action, providing additional data about other actions (i.e., those not selected by the algorithm) will not influence the algorithm's decision.

**Definition 3.1** (Independence of irrelevant data). *A deterministic base algorithm $\Pi$ satisfies the* **independence of irrelevant data** *(IIData) property if whenever $A = \Pi(\mathcal{H})$ then*

$$\Pi(\mathcal{H}) = \Pi(\mathcal{H} \cup \mathcal{H}') \tag{4}$$

*for any $\mathcal{H}'$ containing data from any actions $a'$ other than $A$ (that is, $a' \neq A$).*

IIData is a natural robustness property for an algorithm to satisfy, highlighting that the algorithm evaluates actions independently when making decisions. IIData is conceptually analogous to the independence of irrelevant alternatives (IIA) axiom in computational social choice as a desiderata used to evaluate voting rules (Arrow, 1951). In Theorem 3.2 we show that for any base algorithm satisfying IIData, the regret of $\pi^{\text{FULL START}(\Pi)}$ and $\pi^{\text{ARTIFICIAL REPLAY}(\Pi)}$ will be equal.

**Theorem 3.2.** *Suppose that algorithm $\Pi$ satisfies the independence of irrelevant data property. Then for any problem instance, horizon $T$, and historical dataset $\mathcal{H}^{hist}$ we have the following:*

$$\pi_t^{\text{ARTIFICIAL REPLAY}(\Pi)} \stackrel{d}{=} \pi_t^{\text{FULL START}(\Pi)}$$

$$\text{REGRET}(T, \pi^{\text{ARTIFICIAL REPLAY}(\Pi)}, \mathcal{H}^{hist}) \stackrel{d}{=} \text{REGRET}(T, \pi^{\text{FULL START}(\Pi)}, \mathcal{H}^{hist}) \,.$$

---

**Algorithm 2** Monotone UCB (MONUCB)

1: Initialize $n_1(a) = 0$, $\overline{\mu}_1(a) = 1$, and $\text{UCB}_1(a) = 1$ for each $a \in [K]$
2: **for** $t = \{1, 2, \ldots\}$ **do**
3:     Let $A_t = \arg\max_{a \in [K]} \text{UCB}_t(a)$
4:     Receive reward $R_t$ sampled from $\Re(A_t)$
5:     Update $n_{t+1}(A_t) = n_t(A_t) + 1$, $n_{t+1}(a) = n_t(a)$ for $a \neq A_t$
6:     Update $\overline{\mu}_{t+1}(A_t) = (n_t(A_t)\overline{\mu}_t(A_t) + R_t)/n_{t+1}(A_t)$, $\overline{\mu}_{t+1}(a) = \overline{\mu}_t(a)$ for $a \neq A_t$
7:     Update $\text{UCB}_{t+1}(a) = \text{UCB}_t(a)$ for $a \neq A_t$ and
       $\text{UCB}_{t+1}(A_t) = \min\{\text{UCB}_t(A_t), \overline{\mu}_{t+1}(A_t) + \sqrt{2\log(T)/n_{t+1}(A_t)}\}$

---

This theorem shows that ARTIFICIAL REPLAY allows us to achieve identical regret guarantees as FULL START while simultaneously using data more efficiently. In the subsequent section, we show three example regret-optimal algorithms which satisfy this property, even in the complex CMAB-CRA setting. The algorithms we modify are all UCB-based algorithms. In fact, it is easy to modify most UCB-based algorithms to satisfy IIData by simply imposing monotonicity of the confidence bound estimates for an action rewards. This is easily implementable and preserves all regret guarantees. While for brevity we only discuss IIData algorithms in the $K$-Armed and CMAB-CRA set-up, it is easy to see how to modify other UCB-based algoriths (e.g. LinUCB for linear bandits (Abbasi-Yadkori et al., 2011)) to satisfy IIData. In the existing bandit literature, there has been a narrow focus on only finding regret-optimal algorithms. We propose that IIData is another desirable property that implies ease and robustness for optimally and efficiently incorporating historical data.

## 4 IIDATA ALGORITHMS

In this section, we provide IIData algorithms with optimal regret guarantees for two settings: the $K$-armed and CMAB-CRA models. We show that IIData is easy to guarantee for UCB algorithms requiring only a minor modification to existing algorithms while not impacting confidence bounds guarantees.

We defer algorithm details to Appendix D and proofs to Appendix F. We'll show in Appendix E that in practice, ARTIFICIAL REPLAY still performs nearly optimally even with algorithms that do not satisfy IIData.

### 4.1 $K$-ARMED BANDITS

The first algorithm we propose, named Monotone UCB (denoted as MONUCB), is derived from the UCB1 algorithm introduced in Auer et al. (2002). At every timestep $t$, the algorithm tracks the following: *(i)* $\overline{\mu}_t(a)$ for the estimated mean reward of action $a \in [K]$, *(ii)* $n_t(a)$ for the number of times the action $a$ has been selected by the algorithm prior to timestep $t$, and *(iii)* $\text{UCB}_t(a)$ for an upper confidence bound estimate for the reward of action $a$. At every timestep $t$, the algorithm picks the action $A_t$ which maximizes $\text{UCB}_t(a)$ (breaking ties deterministically). After observing $R_t$, we increment $n_{t+1}(A_t) = n_t(A_t) + 1$, update $\overline{\mu}_{t+1}(A_t)$, and set:

$$\text{UCB}_{t+1}(A_t) = \min\left\{\text{UCB}_t(A_t), \ \overline{\mu}_{t+1}(A_t) + \sqrt{\tfrac{2\log(T)}{n_{t+1}(A_t)}}\right\}. \tag{5}$$

The only modification of Monotone UCB from standard UCB is the additional step forcing the UCB estimates to be monotone decreasing over $t$. It is clear that this modification has no affect on the regret guarantees. Under the "good event" analysis, if $\text{UCB}_t(a) \geq \mu(a)$ with high probability, then the condition still holds at time $t + 1$, even after observing a new data point. In the following theorem, we show that MONUCB satisfies IIData and is regret-optimal, achieving the same instance-dependent regret bound as the standard UCB algorithm.

**Theorem 4.1.** *Monotone UCB satisfies the IIData property and has for $\Delta(a) = \max_{a'} \mu(a') - \mu(a)$:*

$$\text{REGRET}(T, \pi^{\text{IGNORANT(MONUCB)}}, \mathcal{H}^{hist}) = O(\textstyle\sum_a \log(T)/\Delta(a)). \tag{6}$$

This guarantee allows us to use Theorem 3.2 to establish that ARTIFICIAL REPLAY and FULL START have identical regret with MONUCB as a base algorithm. In the next theorem, we

show that ARTIFICIAL REPLAY is robust to *spurious data*, where the historical data has excess samples $a_j^{\mathcal{H}}$ coming from poor performing actions. Spurious data imposes computational challenges, since the FULL START approach will pre-process the full historical dataset regardless of the observed rewards or the inherent value of the historical data. In contrast, ARTIFICIAL REPLAY will only use the amount of data useful for learning.

**Theorem 4.2.** *For every $H \in \mathbb{N}$ there exists a historical dataset $\mathcal{H}^{hist}$ with $|\mathcal{H}^{hist}| = H$ where the runtime of $\pi^{\text{FULL START(MONUCB)}} = \Omega(H + T)$ whereas the runtime of $\pi^{\text{ARTIFICIAL REPLAY(MONUCB)}} = O(T + \min\{\sqrt{T}, \log(T)/\min_a \Delta(a)^2\})$.*

This highlights that the computational overhead of ARTIFICIAL REPLAY in comparison to FULL START can be arbitrarily better. For storage requirements, the FULL START algorithm requires $O(K)$ storage for maintaining estimates for each arm. In contrast, a naive implementation of ARTIFICIAL REPLAY requires $O(K + H)$ storage since the entire historical dataset needs to be stored. However, using hashing techniques can address the extra $H$ factor. In Section 4.2 we will see an example where IIData additionally has strong storage benefits over FULL START.

Lastly, to complement the computational improvements of ARTIFICIAL REPLAY applied to MONUCB, we can also show an improvement of regret. This analysis crucially uses the regret coupling, since FULL START(MONUCB) is much easier to reason about than its ARTIFICIAL REPLAY counterpart.

**Theorem 4.3.** *Let $H_a$ be the number of datapoints in $\mathcal{H}^{hist}$ for each action $a \in [K]$. Then the regret of Monotone UCB with historical dataset $\mathcal{H}^{hist}$ is:*

$$\text{REGRET}(T, \pi^{\text{ARTIFICIAL REPLAY(MONUCB)}}, \mathcal{H}^{hist}) \leq O\Big( \sum_{a \in [K]: \Delta_a \neq 0} \max\big\{0, \tfrac{\log(T)}{\Delta(a)} - H_a \Delta(a)\big\}\Big).$$

Theorem 4.2 together with Theorem 4.3 helps highlight the advantage of using ARTIFICIAL REPLAY over FULL START in terms of improving computational complexity while maintaining an equally improved regret guarantee. This reduces to the standard UCB guarantee when $\mathcal{H}^{hist} = \emptyset$. Moreover, it highlights the impact historical data can have on the regret. If $|H_a| \geq \log(T)/\Delta(a)^2$ for each $a$ then the regret of the algorithm will be constant not scaling with $T$. We note that there are no existing regret lower bounds for incorporating historical data in bandit algorithms. Our main goal is not to improve regret guarantees (although Theorem 4.3 highlights the advantage of historical data), but instead highlight a simple, intuitive, and implementable approach through ARTIFICIAL REPLAY which matches the performance of FULL START while simultaneously having smaller compute.

We close with an example of a $K$-armed bandit algorithm which does not satisfy the IIData assumption. Thompson Sampling (Russo et al., 2018), which samples arms according to the posterior probability that they are optimal, does not satisfy IIData. Data from other actions other than the one chosen will adjust the posterior distribution, and hence will adjust the selection probabilities as well. While we do not obtain a regret coupling, in Fig. 8 (appendix) we show that there are still empirical gains for using ARTIFICIAL REPLAY over FULL START across a variety of base algorithms.

## 4.2 CMAB-CRA

Incorporating historical data optimally and efficiently is difficult in continuous action settings. Two natural approaches are to *(i)* discretize the action space $\mathcal{A}$ based on the data using nearest neighbor estimates, or *(ii)* learn a regression of the mean reward using available data. Consider a setting where excessive data is collected from poor-performing actions. Discretization-based algorithms will unnecessarily process and store a large number of discretizations in low-performing regions of the space. Regression-based methods will use compute resources to learn an accurate predictor of the mean reward in irrelevant regions. The key issues are that the computational and storage cost grows with the size of the historical dataset, and the estimation and discretization is done independent of the quality of the reward.

To contrast this approach, we present two discretization-based algorithms that satisfy IIData with strong performance guarantees. In particular, we detail fixed and adaptive discretization (ADAMONUCB in Algorithm 3) algorithms that only use the historical dataset to update estimates

of the reward. Due to space, we describe the algorithms only at a high level and defer details to Appendix D.

Our algorithms are Upper Confidence Bound (UCB) style as the selection rule maximizes Eq. (1) over the combinatorial action set (Eq. (3)) through a discretization of $\mathcal{S}$. For each allocation $\beta \in \mathcal{B}$, the algorithm maintains a collection of regions $\mathcal{P}_t^\beta$ of $\mathcal{S}$ which covers $\mathcal{S}$. For the fixed discretization variant, $\mathcal{P}_t^\beta$ is fixed at the start of learning, and in the adaptive discretization version it is refined over the course of learning based on observed data. At every timestep $t$ and region $\mathcal{R} \in \mathcal{P}_t^\beta$, the algorithm tracks the following: *(i)* $\overline{\mu}_t(\mathcal{R}, \beta)$ for the estimated mean reward of region $\mathcal{R}$ at allocation $\beta$, *(ii)* $n_t(\mathcal{R}, \beta)$ for the number of times $\mathcal{R}$ has been selected at allocation $\beta$ prior to timestep $t$, and *(iii)* $\text{UCB}_t(\mathcal{R}, \beta)$ for an upper confidence bound estimate. At a high level, our algorithm performs three steps in each iteration $t$:

1. **Action selection**: Greedily select at most $N$ regions in $\mathcal{P}_t^\beta$ to maximize $\text{UCB}_t(\mathcal{R}, \beta)$ subject to the budget constraints (see Eq. (10) in the appendix). Note that we must additionally ensure that each region is selected at only a *single* allocation value.
2. **Update parameters**: For each of the selected regions, increment $n_t(\mathcal{R}, \beta)$ by one, update $\overline{\mu}_t(\mathcal{R}, \beta)$ based on observed data, and set $\text{UCB}_{t+1}(\mathcal{R}, \beta) = \min\{\text{UCB}_t(\mathcal{R}, \beta), \overline{\mu}_t(\mathcal{R}, \beta) + b(n_t(\mathcal{R}, \beta))\}$ for some appropriate bonus term $b(\cdot)$. This enforces monotonicity in the UCB estimates similar to MONUCB and is required for the IIData property.
3. **Re-partition**: This step differentiates the *adaptive discretization* algorithm from fixed discretization, which maintains the same partition across all timesteps. We split a region when the confidence in its estimate (i.e., $b(n_t(\mathcal{R}, \beta))$) is smaller than the diameter of the region. This condition may seem independent of the *quality* of a region, but since it is incorporated into a learning algorithm, the number of samples in a region is correlated with its reward. In Fig. 4 (appendix) we highlight how the adaptive discretization algorithm hones in on regions with large reward without knowing the reward function before learning.

These algorithms modify existing approaches applied to CMAB-CRA in the bandit and reinforcement learning literature, which have been shown to be regret-optimal (Xu et al., 2021; Sinclair et al., 2021). We additionally note that these approaches are IIData.

**Theorem 4.4.** *The fixed and adaptive discretization algorithms when using a "greedy" solution to solve Eq. (1) have property IIData.*

Here we require the algorithm to use the standard "greedy approximation" to Eq. (1), which is a knapsack problem in the CMAB-CRA set-up (Williamson & Shmoys, 2011). This introduces additional approximation ratio limitations in general. However, under additional assumptions on the mean reward function $\mu(\mathbf{p}, \beta)$, the greedy solution is provably optimal. For example, optimality of the greedy approximation holds when $\mu(\mathbf{p}, \beta)$ is piecewise linear and monotone, or more broadly when $\mu(a)$ is submodular. See Appendix D for more discussion.

Finally, we comment that the FULL START implementation of these adaptive discretization algorithms will have storage and computational costs proportional to the size of the historical dataset (since the algorithms ensure that the discretization scales with respect to the number of samples). In contrast, ARTIFICIAL REPLAY uses only a fraction of the historical dataset and so again has improved computation and storage complexity. This is validated in the experimental results in Appendix E.

## 5 EXPERIMENTS

We show the benefits of ARTIFICIAL REPLAY by showing that our meta-algorithm achieves identical performance to FULL START while offering significant practical advantages in reducing runtime and storage. We consider two classes of bandit domains: $K$-armed and CMAB-CRA. As part of our evaluation on combinatorial bandits, we introduce a new model for green security games with continuous actions by adaptively discretizing the landscape of a large protected area of Murchison Falls National Park in Uganda.

All of the code to reproduce the experiments is available at `https://github.com/lily-x/artificial-replay`. Results are averaged over 60 iterations with random seeds, with standard error plotted; experiment details and additional results are available in Appendix E.

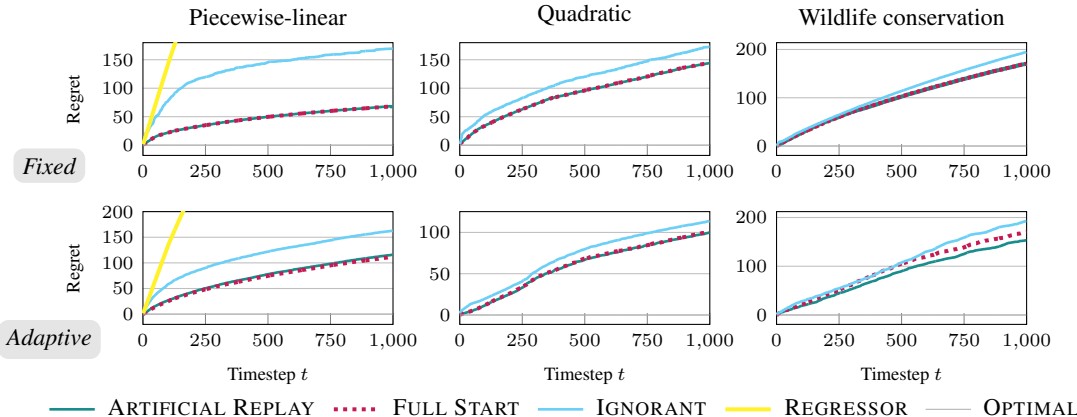

Figure 1: *(CMAB-CRA)* Cumulative regret ($y$-axis; lower is better) across time $t \in [T]$. ARTIFICIAL REPLAY performs equally as well as FULL START across all domain settings, including both fixed discretization (top row) and adaptive discretization (bottom). REGRESSOR performs quite poorly.

**Domains.** We conduct experiments on the two bandit models described in Section 2.2: finite $K$-armed bandits and CMAB-CRA, using both fixed and adaptive discretization. For the continuous combinatorial setting, we provide two stylized domains: a piecewise-linear and a quadratic reward function. To emphasize the practical benefit of ARTIFICIAL REPLAY, we evaluate on a real-world resource allocation setting for biodiversity conservation. We study real ranger patrol data from Murchison Falls National Park, shared as part of a collaboration with the Uganda Wildlife Authority and the Wildlife Conservation Society. We use historical patrol observations to build the history $\mathcal{H}^{hist}$; we analyze these historical observations in detail in Appendix E to show that this dataset exhibits both spurious data and imbalanced coverage as discussed in Section 4.

**Baselines.** We compare ARTIFICIAL REPLAY against IGNORANT and FULL START approaches for each setting. In the $K$-armed model, we use MONUCB as the base algorithm. In CMAB-CRA we use fixed and adaptive discretization as well as REGRESSOR, a neural network learner that is a regression-based approach analogue to FULL START. REGRESSOR is initially trained on the entire historical dataset, then iteratively retrained after 128 new samples are collected. We also compute for each setting the performance of an OPTIMAL action based on the true rewards and a RANDOM baseline that acts randomly while satisfying the budget constraint.

**Results.** The results in Fig. 1 empirically validate our theoretical result from Theorem 3.2: the performance of ARTIFICIAL REPLAY is identical to that of FULL START, and reduces regret considerably compared to the naive IGNORANT approach. We evaluate the regret (compared to OPTIMAL) of each approach across time $t \in [T]$. Concretely, we consider the three domains of piecewise-linear reward, quadratic reward, and green security with continuous space $\mathcal{S} = [0, 1]^2$, $N = 5$ possible action components, a budget $B = 2$, and 3 levels of effort. We include $H = 300$ historical data points. See Fig. 9 (appendix) for regret and analysis of historical data use on the $K$-armed bandit.

Not only does ARTIFICIAL REPLAY achieve equal performance, but its computational benefits over FULL START are clear even on practical problem sizes. As we increase historical data from $H = \{10; 100; 1,000; 10,000\}$ in Fig. 2, the proportion of irrelevant data increases. Our method achieves equal performance, overcoming the previously unresolved challenge of *spurious data*, while FULL START suffers from arbitrarily worse storage complexity (Theorem 4.2). With 10,000 historical samples and a time horizon of 1,000, we see that 58.2% of historical samples are irrelevant to producing the most effective policy.

When faced with *imbalanced data coverage*, the benefits of ARTIFICIAL REPLAY become clear—most notably in the continuous action setting with adaptive discretization. In Fig. 3, as we increase the number of historical samples on bad regions (bottom 20th percentile of reward), the additional data require finer discretization, leading to arbitrarily worse storage and computational complexity for FULL START with equal regret. In Fig. 3(c), we see that with 10% of data on bad arms, AR-

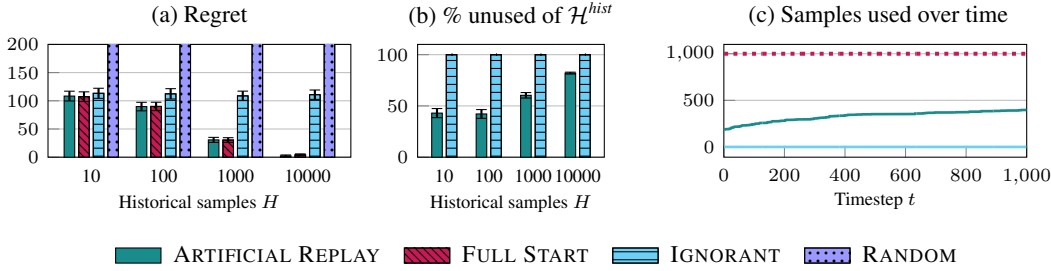

Figure 2: ($K$-Armed) Increasing the number of historical samples $H$ leads FULL START to use unnecessary data, particularly as $H$ gets very large. ARTIFICIAL REPLAY achieves equal performance in terms of regret (plot a) while using less than half the historical data (plot b). In plot c we see that with $H = 1,000$ historical samples, ARTIFICIAL REPLAY uses (on average) 117 historical samples before taking its first online action. The number of historical samples used increases at a decreasing rate, using only 396 of $1,000$ total samples by the horizon $T$. Results are shown on the $K$-armed bandit setting with $K = 10$ and horizon $T = 1,000$.

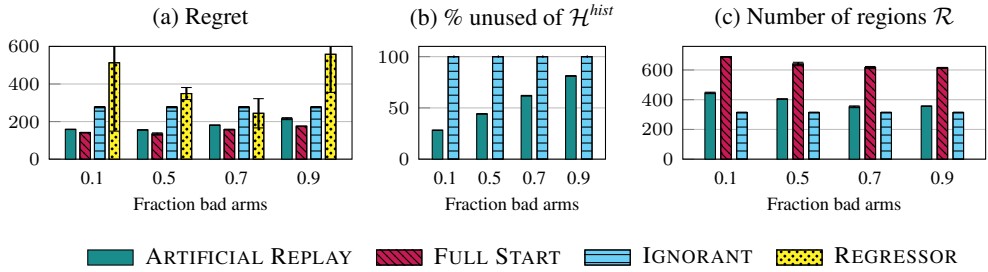

Figure 3: (CMAB-CRA) Holding $H = 10,000$ constant, we increase the fraction of historical data samples on bad arms (bottom 20% of rewards). The plots show (a) regret, (b) % of unused historical data and (c) number of discretized regions in partition $\mathcal{P}$. ARTIFICIAL REPLAY enables significantly improved runtime and reduced storage while matching the performance of FULL START. Results on the CMAB-CRA setting with adaptive discretization on the quadratic domain.

TIFICIAL REPLAY requires only 446 regions $\mathcal{R}$ compared to 688 used by FULL START; as we get more spurious data and that fraction increases to 90%, then ARTIFICIAL REPLAY requires only 356 regions while FULL START still stores 614 regions.

## 6  CONCLUSION

We present ARTIFICIAL REPLAY, a meta-algorithm that modifies *any base bandit algorithm* to efficiently harness historical data. We show that under a widely applicable IIData condition, the regret of ARTIFICIAL REPLAY (as a random variable) is distributionally identical to that of a full warm-start approach, while also guaranteeing significantly better time complexity. We additionally give examples of regret-optimal IIData algorithms in the $K$-armed and CMAB-CRA settings. Our experimental results highlight the advantage of using ARTIFICIAL REPLAY over FULL START via a variety of base algorithms, applied to $K$-armed and continuous combinatorial bandit models, including for algorithms such as Thompson sampling and Information Directed Sampling (IDS) that do not exhibit IIData. Directions for future work include *(i)* find IIData algorithms in other bandit domains such as linear contextual bandits, *(ii)* incorporate the ARTIFICIAL REPLAY approach into reinforcement learning, and *(iii)* provide theoretical bounds showing that ARTIFICIAL REPLAY has *optimal* data usage when incorporating historical data.

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
