# OpenReview forum: "Artificial Replay: A Meta-Algorithm for Harnessing Historical Data in Bandits"
_ICLR.cc/2023/Conference — Submitted to ICLR 2023_

### Official Review · Reviewer_ZUbC · 2022-10-22

**Confidence:** 3
**Clarity, Quality, Novelty And Reproducibility:** Overall this paper is good but the th…
**Correctness:** 3
**Technical Novelty And Significance:** 3
**Empirical Novelty And Significance:** 3
**Recommendation:** 5

**Strength And Weaknesses:**

The problem this paper studied is important and interesting. The meta-algorithm is simple and intuitive. I feel before we talk about computational efficiency and storage, it is better to fully understand the statistical efficiency (regret) for this problem. I have some questions about the regret guarantee and use Theorem 4.3 as an example since this is the basic multi-armed bandits.

1. I feel there lacks sufficient discussion on how the historical data can reduce the online regret. The paper just mentioned "equally improved regret guarantee". This should be discussed quantitively. How about minimax regret? The abstract mentioned spurious and imbalance data could appear in historical data. How they affect the regret?

2. I feel different historical data could affect online regret in a very different way. For example, if historical data contains all the optimal actions, online algorithms could just do imitation learning. If historical data contains a lot very sub-optimal actions, we could use them to remove bad actions. This work seems to not cover those issues. And the question is what exactly a full start algorithm is? There are many different ways to use full historical data and this also depends on what is your base algorithm.

3. In (2), the regret definition seems to be independent of historical data. How does this happens? Should it be conditional on historical data? Will a_j^H be random variables?

4. The authors claimed they propose regret-optimal IIData policies. Where the optimality comes from? I didn't see any lower bound here and is this instance-dependent or minimax optimal?






**Summary Of The Paper:**

This paper studied an important bandit problem: how to deal with historical data. This paper proposed a meta-algorithm to handle computation and storage issues.

**Summary Of The Review:**

Good problem but the theory part needs to be strengthened.

---

> ### Author Response · Authors · 2022-11-19
> **Clarifying regret guarantees and optimality**
>
> We are glad you find the problem important and interesting, and our proposed meta-algorithm intuitive to overcome computational and storage issues.
>
> ## Weakness 1: Clarification on “equally improved regret guarantee”
>
> We have clarified the writing in the main paper around this comment on “equally improved regret guarantee”.  Our main goal of the paper was not to improve regret bounds, but instead to propose a simple algorithm (Artificial Replay) capable of achieving the same regret bounds (as Full Warm Start) with much smaller compute overhead and simpler implementation (seen in Theorem 4.2).  We have additionally added more comments around Theorem 4.3 which highlights the improvement on regret obtained by incorporating historical data.
>
> Spurious and imbalance data are discussed prior to Theorem 4.2, and are used to illustrate the fact that in the presence of additional data on sub-optimal arms the Artificial Replay will have improved _computational requirements_ while still obtaining the same regret improvements as a Full warm start approach.
>
>
> ## Weakness 2: Other ways to incorporate historical data, depending on distribution
>
> There are many ways for incorporating historical data into designing an algorithm.  We agree that if the algorithm had prior knowledge on the _quality_ of the historical data (i.e. all the actions sampled are optimal, or all the actions sampled are sub-optimal) then the approach for incorporating this into an algorithm could be different.  However, in practice, it is not known which regime you are in or how good the historical data is upfront.
>
> Our artificial replay framework works irrespectively on the quality of the historical data, achieving an equal regret improvement to Full Start, while simultaneously being easier to implement and reducing computation overhead.
>
> ## Question: What is a full start algorithm?
>
> We define a “Full Start” algorithm at the top of Section 3: a bandit algorithm that uses the full historical data upfront to initialize the historical estimates.
>
> ## Weakness 3: Regret dependency on historical data
>
> Thanks for this point! We modified the writing and notation for the definition of regret to highlight the dependence of the regret on the historical data.  We view the historical data as fixed, and so $a_j^H$ are not random variables.
>
> ## Weakness 4: Regret optimality of IIData policies
>
> Thanks for this point! We have clarified the writing in the main paper around this comment to clarify optimality of IIData policies.
>
> In saying that we propose “regret-optimal IIData policies”, we refer to the fact that the original algorithms that we are modifying (e.g. UCB, Adaptive Discretization for CMAB-CRA, etc) are “regret-optimal” in the setting without any historical data.  We have additionally added some comments on the fact that there are no existing instance-dependent lower bounds for incorporating historical data into bandit algorithms.  Our main goal of the paper was not to improve regret bounds, but instead to propose a simple algorithm capable of achieving the same regret bounds with much smaller compute overhead and simpler implementation.

---

> > ### Comment · Reviewer_ZUbC · 2022-11-20
> > **Thanks for the response.**
> >
> > Thanks for the response.

---

### Official Review · Reviewer_oZhq · 2022-10-25

**Confidence:** 4
**Clarity, Quality, Novelty And Reproducibility:** Clarity
**Correctness:** 3
**Technical Novelty And Significance:** 3
**Empirical Novelty And Significance:** 3
**Recommendation:** 3

**Strength And Weaknesses:**

Strengths:
1) The problem is very relevant in the context of modern recommendation systems where we often have a lot of historic data. Incorporating the entire historic data in the bandit algorithm can be computationally expensive. So, a natural question that arises is whether we can trade-off a little bit of regret for computational efficiency. The paper takes a step towards answering this question. It shows that under IIData condition, the artificial-replay based algorithm has the same regret as the full-start algorithm while being computationally more efficient.
2) The proposed meta-algorithm is simple and can be used with any bandit algorithm.

Weaknesses:
1) While the IIData condition looks interesting, it is a strong condition. It requires the bandit algorithm's recommendation at a particular time instant to be the same even if it is provided with more information about all the other arms that are not being recommended. None of the bandit algorithms I know of seem to satisfy this condition (e.g., UCB, Thompson Sampling, Phased elimination).  This is worrisome because we are now forced to construct good bandit algorithms that satisfy the IIData condition.  Does this make the decades of work on designing bandit algorithms irrelevant? Is there an easy way to modify any given bandit algorithm to satisfy the IIData condition?

         In practice, there is no need for a stringent requirement that the artificial-replay algorithm and the full start algorithm have the same regret. It is okay to trade-off a little bit of regret for computational efficiency. Is it possible to weaken the IIData condition to take this into account?

2) One important aspect that the paper hasn't touched upon at all is the regret optimality of the proposed algorithms. Is ARTIFICIAL REPLAY(MONUCB) regret optimal for both MAB and CMAB-CRA problems? Are there any regret lower bounds that can be provided in the presence of historic data? These lower bounds can help us understand how optimal the proposed algorithms are.

         Atleast for MAB, it doesn't look like the proposed algorithm is optimal. When there is no historic data, it is well known that standard UCB is not optimal in both minimax and instance dependent sense [1, 2]. There are several other algorithms that have been proposed to fix this issue.

3) At a number of places in the paper, it is claimed that the proposed algorithm is better in computation and storage than the baseline algorithm. However, I don't see this clearly. Note that the full start algorithm only requires storing sufficient statistics (and not the entire data). For MAB, these sufficient statistics are the average reward and number of pulls of each arm. So, if we knew that we are using the full start algorithm ahead of time, then we only need to store these statistics, and the resulting algorithm only requires O(K) storage and O(K) additional compute. This is in fact much better than the artificial replay algorithm which requires O(H) storage and O(sqrt{T}) additional compute. Overall, I believe the MAB setting is not clearly showcasing the computational, storage benefits of the proposed algorithm. A more challenging setting with continuous action spaces (e.g., Linear UCB, Neural UCB) would have been more  interesting. While the authors do consider the CMAB-CRA problem, most of its details are relegated to the appendix. Moreover, the problem is studied by a niche community. I'd instead recommend the authors to consider a more fundamental problem (like LinearUCB) and showcase the benefits of the proposed algorithm.

4) Minor comments:
  (a) the greedy algorithm mentioned in section 4.2 is never defined in the paper.  Given this, it is hard to evaluate the optimality of the proposed algorithm.
  (b) in the statement of theorem 4.2, the run time of artificial-replay algorithm should be the minimum of sqrt{T} of log{T}/square of sub-optimality gap.
  (c) in experiments, it'd be good to report runtime and storage improvements achieved by the proposed algorithm.
  (d) In figure 8 (bottom right), why does the FULL START algorithm have such bad performance? It looks counter intuitive.

[1] Lattimore, Tor. "Optimally confident UCB: Improved regret for finite-armed bandits." arXiv preprint arXiv:1507.07880 (2015).

[2] Garivier, Aurélien, and Olivier Cappé. "The KL-UCB algorithm for bounded stochastic bandits and beyond." In Proceedings of the 24th annual conference on learning theory, pp. 359-376. JMLR Workshop and Conference Proceedings, 2011.

More historic data hurting the regret is weird, counter-intuitive and requires more understanding.

**Summary Of The Paper:**

The paper considers the problem of bandit optimization when the learner has access to historic data. In this setting, the paper studies how best to use the historic data in order to reduce the computational and storage costs of the learner, while not hurting its regret.

The main contribution of the paper is to propose an artificial-replay based algorithm that makes efficient use of the historic data. This algorithm works as a wrapper around any bandit algorithm. For any action recommended by the bandit algorithm, the meta algorithm first checks if there is a data point corresponding to that action in the historic data. If yes, it sends the reward of that action to the learner and removes that point from history. If not, the algorithm simply queries the environment for the reward of the action. Under a condition called "independence of irrelevant data", the authors show that the artificial replay algorithm achieves the same regret as  the regret obtained by warm-starting the bandit algorithm with the entire historic data.

The authors instantiate their framework on two special bandit problems: stochastic multi-armed bandits (MAB), Combinatorial Multi-Armed Bandit for Continuous Resource Allocation (CMAB-CRA). For both these problems, the authors design modified versions of UCB algorithms that satisfy the IIData condition. Experimentally, the authors show that the proposed techniques are computationally more efficient than full warm-start based approaches, which achieving same regret.


**Summary Of The Review:**

While the problem being considered in the paper is interesting, the results could be significantly improved. For example, more interesting examples (than MAB) should be used to illustrate the benefits of the proposed algorithm. Moreover, the paper doesn't really talk about the regret optimality of the proposed algorithms. It'd be great if the authors address my concerns above.

---

> ### Author Response · Authors · 2022-11-18
> **Responding to weaknesses: clarifying IIData, regret optimality, and computational/storage improvements**
>
> Thank you for your thoughtful consideration of our paper. We’re glad to hear that you think the problem is very relevant and our solution widely applicable.
>
> # Weaknesses
>
> ## Weakness 1: Is IIData too strong?
>
> We show in the paper that *it is easy to modify a wide class of bandit algorithms (i.e., UCB-based algorithms) to satisfy IIData* — we provide examples with MonUCB and CMAB-CRA. Note that our monotonicity modification to ensure IIData is easily implementable and preserves all regret guarantees: if the UCB estimate at time $t$ upper bounds the true mean with 95% probability, then that same estimate will also hold at time $t+1$ even if an updated sample increases the UCB.
>
> We have updated the writing in the paper to clarify the applicability of IIData to the broad class of UCB-based algorithms.
>
> ## Question: Trade off between regret for computational efficiency?
>
> There is no tradeoff between regret and computational efficiency: for IIData algorithms we show that Artificial Replay shows the same regret (see Theorem 4.1) but with much smaller computational requirements (see Theorem 4.2 and 4.3).  However, other algorithms not satisfying the IIData property might have a more nuanced relationship between regret and computational performance; the
>
> ## Weakness 2: Regret optimality of proposed algorithm, and shortcomings of standard UCB
>
> We have extended the discussion around Theorem 4.3 to discuss regret optimality of these approaches.  There are no existing regret lower bounds for incorporating historical data in bandit algorithms, which is a strong direction for future work.  Our main goal in the paper was not to improve regret guarantees, but instead highlight a simple, intuitive, and implementable approach for incorporating historical data that matches performance of full start, while simultaneously having smaller compute overhead.
>
> You point out that standard UCB is not optimal in minimax and instance-dependent regret. We agree that the constant term in front of the standard UCB algorithm is not optimal, in contrast to KL-UCB or other variants.  However, the asymptotic gap-dependent performance is indeed optimal.  We will consider how to incorporate the IIData property in KL-UCB for future work. Note that we have already implemented and evaluated the performance of Artificial Replay with Thompson sampling and Information-Directed Sampling in the appendix (see Figure 8), where we still observe equal of Artificial Replay compared to the full warm start, and significantly improved performance over an approach that ignores history.
>
> ## Weakness 3: Computational and storage improvements proposed algorithm
>
> Thank you for pointing out these nuances about the storage requirements of Artificial Replay. We agree that a full discussion is more nuanced; we have incorporated these comments in the revision.
>
> You are correct in that the Full Warm start algorithm requires $O(K)$ storage requirement for maintaining the estimates for each arm.  A naive implementation of the Artificial Replay framework will require $O(K+H)$ storage (for storing the full historical data).  However, additional storage structures (e.g. hashing) could help address these.  We also note that most practical bandit applications incorporate historical data obtained from database systems (e.g. content recommendation systems, wildlife poaching model discussed).  This historical data will be stored regardless of the algorithm being employed, and so the key consideration is around computational requirements and not storage.  We have adjusted the writing in the paper for this fact.
>
> The computational requirements for Artificial Replay, however, are much better than that of Full Warm Start.  While the Full Warm Start algorithm only maintains $K$ statistics, it requires $O(H)$ additional compute because each datapoint needs to be read in order to update and maintain the mean and confidence interval estimates.  In contrast, Artificial Replay only requires $O(log(T))$ additional compute, which is a substantial improvement in practical settings with access to large databases of historical data.
>
> We do not agree that the K-Armed and CMAB-CRA models are niche, since they have practical insights into designing algorithms for wildlife poaching prevention mechanisms as highlighted in the experimental results.  Moreover, the CMAB-CRA domain includes continuous actions as recommended in this comment.  Additionally, the CMAB-CRA domain has additional storage insights since there are growing storage requirements when using adaptive discretization (even when storing summary statistics) with respect to the historical data. See Figure 3(b,c) and Figure 7 (appendix) for additional discussion on this facet.
>
> We have added additional comments on incorporating historical data through LinUCB as another example of IIData algorithms in the appendix.

---

> > ### Comment · Reviewer_oZhq · 2022-11-19
> > **Regarding weaknesses**
> >
> > *  ``We show in the paper that it is easy to modify a wide class of bandit algorithms (i.e., UCB-based algorithms) to satisfy IIData''
> >    - In the paper, I could only see two algorithms being modified to satisfy IIData assumption. Given this, I don't see how the above claim is justified. If the authors want to make a bigger claim, I'd recommend having a concrete theorem proving that all UCB-based algorithms satisfy the assumption.
> >
> > * ``We will consider how to incorporate the IIData property in KL-UCB for future work.''
> >    - this seems to contradict with the above claim that the paper provides a way to modify a wide class of bandit algorithms to satisfy the IIData property.
> >
> > *  ``Our main goal in the paper was not to improve regret guarantees, but instead highlight a simple, intuitive, and implementable approach for incorporating historical data that matches performance of full start, while simultaneously having smaller compute overhead.''
> >    - This would have been perfectly fine if the proposed approach can handle any base algorithm. But that's not the case. The proposed approach can only handle algorithms that satisfy IIData assumption. Given this, I believe it is important to also talk about regret optimality.
> >
> > * ``We agree that the constant term in front of the standard UCB algorithm is not optimal''
> >   -  I'd like to note that UCB for MAB is sub-optimal - in the minimax regret sense - by log factors (not constant factors). Even if we care about  asymptotic gap-dependent bounds, UCB is sub-optimal (the amount of sub-optimality depends on the true distribution). This is because the lower bounds for regret depend on KL divergence, whereas the regret upper bounds of UCB depend on squared distance between the mean of arm rewards.

---

> > > ### Author Response · Authors · 2022-11-19
> > > **Response**
> > >
> > > Thanks for your comments! I think there are a couple confusions in our response, so hopefully this clears it up.
> > >
> > > ### Modifying UCB Algorithms to Satisfy IIData
> > >
> > > We include three example algorithms that satisfy IIData, in the $K$-armed and the CMAB-CRA problem set-ups (Monotonce UCB and fixed and adaptive discretization for the CMAB-CRA problem).  In the updated revision on the top of page six we include a discussion on how it is simple to modify existing UCB-based algorithms to satisfy the property by ensuring that the confidence estimates on the arm rewards are monotone decreasing.  In the Monotone UCB example, this amounts to running the "standard" algorithm, but upon receiving more data from an action simply making sure that after updating mean and confidence estimates that the resulting UCB estimate is snapped to whatever it was previously if it was smaller.  This is easily implementable (and could be done similarly for KL-UCB and LinearUCB) and preserves all regret guarantees: if the UCB estimate at time $t$ upper bounds the true mean with 95% probability, then that same estimate will also hold at time $t+1$ even if an updated sample increases the UCB.
> > > In the revisions we will try and clear up any other confusions in this section, and think about trying to write a unified theorem for looking at UCB-based algorithms to satisfy the property.
> > >
> > > As for the comment re. KL-UCB, was mostly implying that we will modify Figure 8 in the appendix running experiments on $K$-armed bandit algorithms (including Monotone UCB, Thompson Sampling, and Information Directed Sampling) to additionally include KL-UCB as suggested.
> > >
> > > ### (misunderstanding) Artificial Replay requires IIData to Work
> > >
> > > You suggested that Artificial Replay only works when the base algorithm has IIData property. However, Artificial Replay works on top of any base bandit algorithm. Although the regret coupling guarantees (Theorem 4.2) only apply when IIData applies, the approach can be used with any bandit algorithm.
> > >
> > > For example, in Figure 8 of the appendix, we test Artificial Replay with Information-Directed Sampling (IDS) and Thompson Sampling algorithms — two algorithms that do not have IIData property — and we still observed equal regret performance compared to the full warm start, and significantly improved performance over an approach that ignores history.  However, when comparing the computational complexity the Artificial Replay algorithms had substantial improvements.

---

> ### Author Response · Authors · 2022-11-18
> **Responding to minor comments**
>
> ## Minor comments
>
> (a) We have updated writing in Theorem 4.4 to clarify “a greedy solution”, which is that the knapsack problem (equation 10) is optimized in a greedy fashion.  We note in the Appendix that this is a standard approximation algorithm framework for solving knapsack problems, and is typically done in practice.
>
> (b) Thank you for pointing this out to us, we have fixed this in the revision.
>
> (c) We include storage improvements in Figures 2 and 3, as subplot (b) to indicate the percentage unused of the historical dataset. We did look at runtime, but each run that we consider finishes on the order of seconds, so we omitted including those results as the runtime is negligible.
>
> (d) Good point! We have added some clarification around this fact in the revision.   We agree that this is counter-intuitive (and is a facet that is avoided by having algorithms which satisfy the IIData property, since any IIData algorithm cannot have worse regret when incorporating historical data).  Figure 8 is showing the performance of Information Directed Sampling (an algorithm which does not satisfy the IIData property).  What is happening practically in this experiment is the IDS algorithm maintains a near-“zero-entropy” posterior on a sub-optimal action which is over-represented in the historical dataset.  Since the selection rule picks actions which maximize the expected return divided by the posterior variance, the algorithm continuously picks this sub-optimal action at each timestep.

---

### Official Review · Reviewer_NKDb · 2022-10-25

**Confidence:** 4
**Correctness:** 4
**Technical Novelty And Significance:** 3
**Empirical Novelty And Significance:** 3
**Recommendation:** 6

**Clarity, Quality, Novelty And Reproducibility:**

The paper is well-written and is very clear and nicely structured. The key idea is highlighted appropriately and the main algorithm as well as the experiments are presented well. The algorithm itself is very natural and simple - but I actually consider that a positive.

The appendix includes enough details for reproducibility (especially once the code is released).

**Strength And Weaknesses:**

Strengths:
+ Proposed algorithm is simple and intuitive.
+ The IIData condition is novel, yet easy to verify for different algorithms.

Weaknesses:
- Could you add a discussion of gap independent regret? How much data is needed to give asymptotic improvements in the regret?
- It is unclear what the exact computational /  storage costs are for using all the available historical data. For the K-armed bandit problem, the storage costs are simply a function of the number of arms and remain unchanged whether the algorithm builds UCB estimates for all the arms using all the historical data at time 0 or whether it uses the historical data incrementally. Even for the CMAB-CRA problem, the entire available historical data still needs to be stored (for potential future look-ups); so it’s not fully clear what the savings are.
- I am interested in a discussion of the setting where the historical data is also obtained via actions of a no-regret bandit algorithm (which is likely to be the case when such an algorithm is actually deployed). In this setting, the collected historical data is unlikely to contain spurious data - and it will be interesting to see whether the empirical gains still hold.


**Summary Of The Paper:**

The paper considers the stochastic bandit learning problem when the algorithm has access to information regarding historical actions and their corresponding rewards. By leveraging already available information regarding the rewards from historical actions, a bandit algorithm can obtain significantly improved regret over vanilla bandit algorithms. A simple, but naive, approach to use historical information is to incorporate all available data to “initialize” a bandit algorithm - while this approach effectively utilizes the available historical information, it suffers from being computationally expensive. Instead the paper proposes a new meta-algorithm: for any online bandit algorithm, when the bandit algorithm recommends action A_t at time t, instead of actually playing action A_t it checks if action A_t is available in the historical data and updates the internal state of the algorithm using the historical reward instead. The authors show that under certain assumptions on the base bandit algorithm, this strategy has regret equivalent to the naive approach above that uses the entire historical data upfront.

**Summary Of The Review:**

The paper is well-written and proposes a simple, intuitive (meta)-algorithm to harness historical information in stochastic bandit settings. The proposed approach makes efficient use of the available historical information and obtains significantly improved regret without using the full historical dataset at the beginning.

---

> ### Author Response · Authors · 2022-11-18
> **Clarifying "gap-independent regret" and improvements in computation and storage costs**
>
> Thank you for your positive assessment of our paper. We are glad you find our proposed approach intuitive, novel, and well-presented.
>
> ## Weakness 1: Gap independent regret
>
> We have added to the discussion around the gap-independent regret analysis in Theorem 4.3 to highlight historical data requirements for asymptotic improvement.  However, as $T$ approaches infinity, since the historical data is fixed, eventually the asymptotic performance is identical.
>
> ## Weakness 2: Computational / storage costs
>
> We agree that a full discussion around the storage requirements of the algorithm is more nuanced than was previously discussed in the paper, and we have incorporated these comments in the revision.  You are correct in that the Full Warm start algorithm requires $O(K)$ storage requirement for maintaining the estimates for each arm.  A naive implementation of the Artificial Replay framework will require $O(K+H)$ storage (for storing the full historical data).  However, additional storage structures (e.g. hashing) could help address these.  We also note that most practical bandit applications incorporate historical data obtained from database systems (e.g. content recommendation systems, wildlife poaching model discussed).  This historical data will be stored regardless of the algorithm being employed, and so the key consideration is around computational requirements and not storage.  We have adjusted the writing in the paper for this fact.
>
> ## Weakness 3: Historical data obtained from bandit algorithm
>
> The analysis is relatively simple in the setting where the historical data is obtained by the same bandit algorithm which is actually deployed.  You are able to show that $R(T) <= R(T+N) - N \Delta_{\min}$, and since $R(T+N)$ will be (hopefully) sublinear in $N$ this will directly show the improvement of regret for incorporating historical data.  Moreover, potentially using a best-arm identification algorithm will be better theoretically for minimax performance in terms of incorporating historical data.  However, our goal was not to show regret improvements, but instead propose a simple approach for incorporating historical data with identical regret to full warm start (approach done in practice) with smaller computational overhead.

---

### Official Review · Reviewer_Ai7i · 2022-10-27

**Confidence:** 4
**Correctness:** 3
**Technical Novelty And Significance:** 2
**Empirical Novelty And Significance:** Not applicable
**Recommendation:** 3

**Clarity, Quality, Novelty And Reproducibility:**

**Clarity:**
The paper is well organized, but the presentation has minor details that could be improved, as discussed earlier in **Strength And Weaknesses**.

**Quality:**
Overall, the paper appears to be technically sound. The proofs appear correct, but I have not carefully checked the details. The experimental evaluation is adequate and supports the main claims.

**Novelty:**
This paper contributes some new ideas, but they only represent incremental advances.

**Reproducibility:**
The key resources (e.g., proofs and code) are available, and sufficient details are given to reproduce the main results.

**Details Of Ethics Concerns:**

Since this work is a theoretical paper, I do not find any ethical concerns.

**Strength And Weaknesses:**

**Strengths of paper:**
1. The problem of exploiting historical data is interesting, and as discussed in the paper, many real-world applications already have historical data.

2. The proposed meta-algorithm only uses a subset of historical data and has the better regret as a bandit algorithm (satisfying IIData property) that uses all historical data for initializing reward estimates.

3. The proposed two-base bandit algorithms, MONUCB (for K-armed bandits) and CMAB-CRA (combinatorial MAB for Continuous Resource Allocation), enjoy IIData property. The authors empirically validated the different performance aspects of the proposed algorithm on synthetic and real datasets.

**Weakness of paper:**
1. The assumption of IIData, i.e., having additional information about other actions won't change the bandit algorithm decision in a given round, is too strong and counterintuitive. Best of my knowledge, no existing bandit algorithm that has IIData property.

2. The statement, "We present ARTIFICIAL REPLAY, a meta-algorithm that modifies any base bandit algorithm to efficiently harness historical data." in the Conclusion and a similar statement in the Abstract is misleading. Because ARTIFICIAL REPLAY only works when the base algorithm has IIData property.

3. The storage and computational problems are not that big in many real-world applications, and the authors have not motivated enough by giving suitable examples of where one should care about these issues. Even for the proposed algorithm, all history needed to be stored, so the storage issue is still there. However, one can have a computationally efficient implementation for accessing historical data. This problem will get more challenging for the continuous action space as finding action within the $\epsilon$ range may be computationally intensive.

4. The proposed algorithm is horizon dependent and needs $T$ as input. Making such an assumption may not be practical in many real-world applications as $T$ (how long the algorithm will be used in practice) may not be known.

5. In the experiments, authors have not used Thomson sampling variants (which enjoy better empirical performance than their UCB counterparts) against their proposed algorithms. It would be interesting to see how MONUCB (with ARTIFICIAL REPLAY) performs against the Thomson sampling variant that does not use any history (even for a K-armed bandit).

6. Some related work for Combinatorial Multi-Armed Bandit for Continuous Resource Allocation is missing, e.g.,
i. Tor Lattimore, Koby Crammer, and Csaba Szepesvári. Optimal resource allocation with semi-bandit feedback. UAI, 2014.
ii. Tor Lattimore, Koby Crammer, and Csaba Szepesvári. Linear multi-resource allocation with semi-bandit feedback. NIPS, 2015.
iii. Yuval Dagan and Crammer Koby. A better resource allocation algorithm with semi-bandit feedback. ALT, 2018.
iv. Other recent work.


**Question and other comments.**

Please address the above weakness. I have one more question:
1. Page 3, paragraph after Eq. (3): Why do chosen resources for different arms needs to be $\epsilon$-away from each other?


I have a few minor comments:
1. $H_1$ needs to be initialized as an empty set.
2. $H^{\text{hist}}$ needs to be updated after getting a sample from it; otherwise, it is possible to sample the same action-reward tuple when the existing action is chosen again.

I am open to changing my score based on the authors' responses.

**Summary Of The Paper:**

This paper studies the stochastic bandits problem, where historical data is available. The goal is to learn a policy that exploits the available historical data to reduce regret (the difference between the maximum achievable reward and the policy's reward).


The authors propose a meta-algorithm named ARTIFICIAL REPLAY that uses a bandit algorithm (satisfying IIData property) as a base algorithm. The proposed algorithm only uses a subset of the historical data (hence reducing the computational cost), but it has the same regret bound as the algorithm that uses all historical data upfront (without suffering from issues like spurious data). They also validate the proposed algorithm's performance on synthetic and real-world datasets.

**Summary Of The Review:**

This paper significantly overlaps with my current work, and I am very knowledgeable about most of the topics covered by the paper.

---

> ### Author Response · Authors · 2022-11-18
> **Clarifying misunderstandings of IIData and applicability of our Artificial Replay algorithm**
>
> Thank you for your detailed assessment of our paper. We are glad to hear that you think the problem we’re addressing is interesting and the theoretical and empirical results valuable.
>
> We respond to each of your stated weaknesses and questions, below, and have also updated our paper draft accordingly.
>
> # Weaknesses
>
> ## (misunderstanding) Weakness 1: Assumption of IIData is too strong
>
> We respectfully point out a misunderstanding.
>
> You claim “no existing bandit algorithm has IIData property.” However, we show in the paper that *it is easy to modify a wide class of bandit algorithms (i.e., UCB-based algorithms) to satisfy IIData* — we provide examples with MonUCB and CMAB-CRA. Note that our monotonicity modification to ensure IIData is easily implementable and preserves all regret guarantees: if the UCB estimate at time $t$ upper bounds the true mean with 95% probability, then that same estimate will also hold at time $t+1$ even if an updated sample increases the UCB.
>
> We have updated the writing in the paper to clarify the applicability of IIData to the broad class of UCB-based algorithms.
>
> ## (misunderstanding) Weakness 2: Artificial Replay
>
> We respectfully point out a second misunderstanding.
>
> You claim that Artificial Replay only works when the base algorithm has IIData property. However, *Artificial Replay works on top of any base bandit algorithm*. Although the regret coupling guarantees (Theorem 4.2) only apply when IIData applies, the approach can be used with any bandit algorithm.
>
> For example, in Figure 8 of the appendix, we test Artificial Replay with Information-Directed Sampling (IDS) and Thompson Sampling algorithms — two algorithms that do not have IIData property — and we still observed equal performance compared to the full warm start, and significantly improved performance over an approach that ignores history.
>
> ## Weakness 3: Storage and computational barriers
>
> We agree that a full discussion around the storage requirements of the algorithm is more nuanced than was previously discussed in the paper, and we have incorporated these comments in the revision.
>
> You are correct in that the Full Warm start algorithm requires $O(K)$ storage requirement for maintaining the estimates for each arm. A naive implementation of the Artificial Replay framework with $K$ arms and $H$ historical data points will require $O(K+H)$ storage  (for storing the full historical data). However, additional storage structures (e.g. hashing) could help address these.
>
> We also note that most practical bandit applications incorporate historical data obtained from database systems (e.g. content recommendation systems, wildlife poaching model discussed).  This historical data will be stored regardless of the algorithm being employed, and so the key consideration is around computational requirements and not storage.  We have adjusted the writing in the paper for this fact.
>
> ## Weakness 4: Algorithm’s dependence on horizon $T$
>
> Thank you for pointing this out. We have updated the writing to remove this dependence on the horizon $T$ (see changes to Algorithm 1).  This is not required for the Artificial Replay meta-algorithm.
>
> ## (misunderstanding) Weakness 5: Comparison to Thompson sampling
>
> Thank you for your recommendation to study Thompson sampling — we have this exact experiment in Figure 8 in the appendix! We compare to Thompson sampling and Information-Directed-Sampling (IDS), where Artificial Replay performs equally well as a “full start” approach (and significantly better than TS with no history).
>
> ## Weakness 6: Related work on CMAB-CRA
>
> Thank you for your pointer to these papers on scheduling problems that depend on level of effort. We have added to our discussion of these papers our introduction to CMAB-CRA in Section 2.2.
>
>
> # Questions and Comments
>
> # Question 1: Requirement for chosen arms to be $\epsilon$ away from each other
>
> Thanks for this question — we have adjusted the discussion to highlight this requirement.  This is needed for practical consideration, as without it, the optimal action could be to just send high effort (at larger capacity one) at a specific location. In the poaching variant this is practically meaningless, and so we included this requirement that the effort allocations must be sufficiently far for patrol locations.
>
> # Comment 1: Initializing $H_1$ and updating $H_\text{hist}$
>
> Thanks for this comment, we have adjusted this in the revision.

---

> > ### Comment · Reviewer_Ai7i · 2022-11-25
> > **Thank you**
> >
> > Hi,
> >
> > Thank you for your detailed responses and clarifications. The IIDdata (i.e., equivalent to saying that having additional information about other actions won't change the bandit algorithm decision in a given round) is a strong assumption for deriving regret bounds. Therefore, I am keeping my score as it is.

---

> > > ### Author Response · Authors · 2022-11-25
> > > **Strength of IIData Assumption**
> > >
> > > We agree that the IIData property is not true for all algorithms.  However, as we have demonstrated in the paper and review discussion: **every setting we know of in online learning admits UCB based algorithms which can be modified to satisfy IIData and are still regret optimal**.  This includes Finite Arm Bandits, Linear Bandits, Contextual Bandits, RL, Lipschitz Bandits, and Bandits with Resource Constraints.
> > >
> > > It would be helpful if you could give us a setting where one can not match optimal regret bounds via an algorithm satisfying the IIData property. Otherwise, the strength of the IIData assumption seems more a matter of misunderstanding.

---

### Decision · Program_Chairs · 2023-01-20

**Decision:**

Reject

**Justification For Why Not Higher Score:**

The assumptions in the analysis are not satisfied by any existing bandit algorithm.

**Justification For Why Not Lower Score:**

N/A

**Metareview: Summary, Strengths And Weaknesses:**

This paper proposes a meta-algorithm for incorporating historical data into any base bandit algorithm. The proposed approach is analyzed and also empirically evaluated.

The initial reviews of the paper were 2x borderline and 2x reject, and this did not change after the discussion. The main issue that leads to low scores is a strong i.i.d. assumption in the analysis, which is not satisfied by any existing bandit algorithm. This is also the main reason for rejection. I wanted to add two more comments:

* There are closely related prior works that are not cited, such as [Warm-starting Contextual Bandits: Robustly Combining Supervised and Bandit Feedback](https://proceedings.mlr.press/v97/zhang19b.html).

* The problem of warm-starting can be naturally treated from the Bayesian point of view, where the prior is learned from historical data using either frequentist or Bayesian techniques. Why would this be a bad idea?